# IMPROVING CLASSIFIER DECISION BOUNDARIES US-ING NEAREST NEIGHBORS

## ABSTRACT

Neural networks are not learning optimal decision boundaries. We show that decision boundaries are situated in areas of low training data density. They are impacted by few training samples which can easily lead to overfitting. We provide a simple algorithm performing a weighted average of the prediction of a sample and its nearest neighbors' (computed in latent space) leading to a minor favorable outcomes for a variety of important measures for neural networks. In our evaluation, we employ various self-trained and pre-trained convolutional neural networks to show that our approach improves (i) resistance to label noise, (ii) robustness against adversarial attacks, (iii) classification accuracy, and to some degree even (iv) interpretability. While improvements are not necessarily large in all four areas, our approach is conceptually simple, i.e., improvements come without any modification to network architecture, training procedure or dataset. Furthermore, they are in stark contrast to prior works that often require trade-offs among the four objectives or provide valuable, but non-actionable insights.

## 1 INTRODUCTION

In the realm of machine learning, the decision boundary plays a crucial role in distinguishing between classes. Classes typically share certain characteristics and tend to form clusters. The decision boundary is a hypersurface that partitions the input space into regions corresponding to different classes. An optimal decision boundary implies an optimal classifier and vice versa. Simple classifiers, such as support vector machines (SVMs), logistic regression, and k-nearest neighbors, often generate decision boundaries, which can be linear or non-linear Cortes & Vapnik (1995), but are overall rather simple. On the other hand, deep learning models, especially deep neural networks (DNNs), have shown remarkable capability in learning feature hierarchies and capturing complex, non-linear decision boundaries owing to their architectural depth and non-linear activation functions LeCun et al. (2015). They can be said to have revolutionized the field of computer vision and others, leading to astonishing improvements in accuracy on multiple benchmarks. Still, these models also suffer from weaknesses such as a lack of interpretability, lack of robustness as witnessed by the effectiveness of adversarial samplesGoodfellow et al. (2014). Many techniques that tackle the problem of adversarial samples, interpretability, as well as improving handling of noisy labels come with tradeoffs. That is, any of these goals often leads to lower accuracy or requires altering training schemes, datasets, and architectures.

Table 1: Comparison to (some) prior work

| Method | Improvements in | | | | Can use pretrained networks? | Better understanding of dec. boun.? |
|---|---|---|---|---|---|---|
| | Accur-acy | Interpr-etabili. | Advers. Robust. | Label Noise Robust. | | |
| Wu et al. (2020) | | | | ✓ | | ✓ |
| Ortiz-Jimenez et al. (2020) | | | | | | ✓ |
| Karimi et al. (2019) | | | | | | ✓ |
| Yang et al. (2020) | | ✓ | | | | ✓ |
| Oyen et al. (2022) | | | | | | ✓ |
| **ours** | ✓ | ✓ | ✓ | ✓ | ✓ | ✓ |

In this work, we propose a technique that should tackle all of these issues as illustrated in Table 1. We combine the prediction of a pre-trained neural network of a sample to predict and the prediction of the k-nearest neighbors (kNNs). We compute nearest neighbors(NNs) in latent space using layer activations of a (pre-trained) classifier. We calculate a weighted average of the actual prediction and those of NNs as final prediction (see Figure 1). This approach manages to improve robustness against adversarial samples, interpretability, and handling noisy samples without compromising performance of the classifier, i.e., we mostly improve it. It also does not require altering a (pretrained) classifier. However, obtaining kNNs is also computationally expensive and our technique does also not provide a full mitigation against any of the issues or desiderata (in Table 1), but rather addresses them partially. This, is still a major step forward given that other techniques require rather unpleasant trade-offs. Our work is also interesting as it sheds new insights on decision boundaries. In our work, we show that classifiers do generally not learn optimal decision boundaries (also) due to the fact that these boundaries lie in areas of few training samples and thus naturally suffer from overfitting. Predictions of samples near the decision boundary, i.e. samples in these sparse areas, can benefit from using NNs. While we expect few (test) data in such sparse areas, differences when altering the boundary using NNs can still be observed.

## 2 METHOD

Our method is different from classical kNNs in three ways. First, we compute nearest neighbors based on a latent representation given by layer activations rather than on input samples. Second, the prediction is a combination of the network output of the sample to predict itself and its NNs, while for classical kNNs only the NNs are used to make a prediction. Third, the combination is based on directly aggregating network outputs rather than performing a majority vote of the classes of the kNNs. The method is illustrated in Figure 1. More formal pseudocode is shown in Algorithm 1 called "LAtent-SElf-kNN"(LaSeNN), since it combines the sample to predict itself and its NNs and computes NNs based on similarity in latent space. We are given a classifier $C = (L_1, ..., L_n)$ consisting of $n$ layers, a training dataset $\mathcal{D} = \{(X, Y)\}$ and a query sample $X_q$ used for inference. We compute the activations $C_{:i}(X)$ of layer $i$ for all samples in the training dataset and the sample for inference, i.e. $X \in \{\mathcal{D} \cup \{X_q\}\}$. Then we compute the $k$-nearest neighbors $NN_k(C_{:i}(X_q)) \subset \mathcal{D}$ and, finally, a weighted average

$$X_w^i = w_q \cdot C(X_q) + (1 - w_q) \cdot \frac{\sum_{X \in NN_k} C(X)}{k}$$

1: **Input:** Classifier $C$, (training) data $\mathcal{D}$, sample to predict $X_q$
2: **Output:** Prediction $Y_p$
3: $k := 3$ {number of NNs}
4: $w_q := 0.88$ {weight of sample $X_q$}
5: $i := n - 1$ {Layer index to obtain embeddings used for similarity computation for NNs}
6: $sim(X, X') := ||X - X'||_2^2$ {similarity metric for NN}
7: $L_D := \{(C_{:i}(X), Y) | (X, Y) \in D\}$
8: $NN_k(C_{:i}(X_q)) := k\text{-NNs of } C_{:i}(X_q) \text{ in } L_D \text{ using metric } sim$
9: $X_w^i = w_q \cdot C(X_q) + (1 - w_q) \cdot \frac{\sum_{X \in NN_k} C(X)}{k}$
10: $Y_p := \arg\max_j X_{w,j}^i$ {Predicted class is index $j$ of "neuron" with maximal output}

**Algorithm 1:** LaSeNN

The underlying motivation is illustrated in Figure 4. Classes form dense clusters that are separated by sparse space. The decision boundary runs through the sparse space. The exact location of the boundary is heavily influenced by the samples in the sparse space and is likely overfitting. Generally, using NNs can lead to a smoother boundary that is simpler and less-prone to overfitting (see e.g. textbooks like Hastie et al. (2009)). Predictions for samples near any of the cluster centers are relatively far from the decision boundary and bear little uncertainty and are likely correct. They are not impacted by our method, i.e., the prediction using Algorithm 1 (LaSeNN) and the prediction of the network without using NNs is identical. However, predictions in the sparse space are potentially

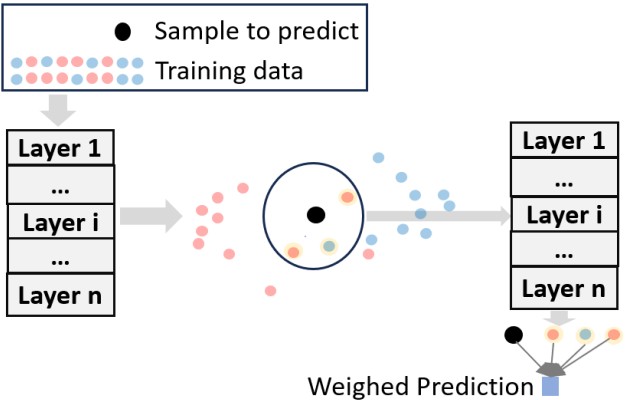

Figure 1: Method: The training data and the sample to infer is run through the classifier up to some layer yielding an embedding, used to compute the kNNs. Finally, a weighted average of the sample to infer and the NNs is used for prediction.

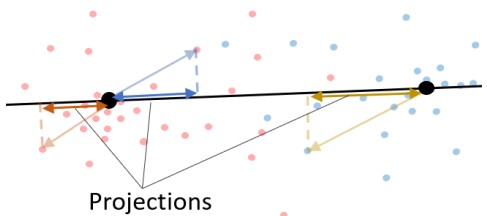

Figure 2: Computation of projections for histograms in Figures 7 and 3. Projections are made onto the line connecting the means of class samples (black dots).

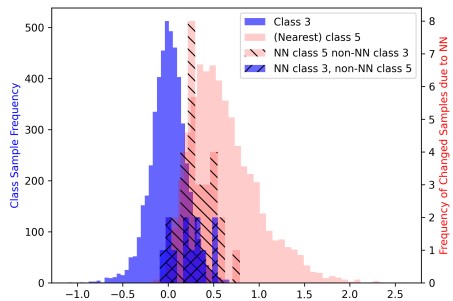

Figure 3: Distribution of projections (see Figure 2) for a VGG-13 trained on Cifar-10 using the third last convolutional layer.

close to the decision boundary, which is strongly influenced by a few samples. Using kNNs leads to changes in this space to the boundary, i.e., a 'novel' decision boundary as illustrated conceptually in Figure 4. (Actual data is shown in Figures 7 and 3.) As shown in Figure 2, for a class $c_0$ we compute the mean of class samples and find the class $c_1$ with the mean that is at minimum L2-distance. We then compute the projection (based on an ordinary dot product) and create histograms. Figure 7 shows that "confusion" of predictions between the two classes occurs primarily in areas of lower density. Figure 3 shows that changes of samples due to Algorithm LaSeNN also occur primarily in low density areas. Furthermore, there are only relatively few changes, e.g., the dense areas containing most samples are not impacted by our method, but only areas of low density. Note that Figure 3 has a twin axis, i.e., the left y-axis is for the distribution of projection of class samples and the right one only for those samples which prediction got changed due to the use of nearest neighbors, i.e. Algorithm LaSeNN. (We also provide some conceptual understanding using more mathematical analysis and abstractions in the Appendix.)

The returned NNs help to better understand classifier decisions, i.e., they are well-interpretable. They indicate which samples of the training data contribute at least the fraction $(1 - w_q)$ to the decision, i.e., each output the classifier of the kNNs has a weight of $\frac{1-w_q}{k}$. Furthermore, in particular, if the layer $i$ is close to the output, the NNs also resemble samples that are considered "very similar" by the classifier and therefore can help in understanding, which concepts are relevant (see concept-based XAI techniques such as Schneider & Vlachos (2022)).

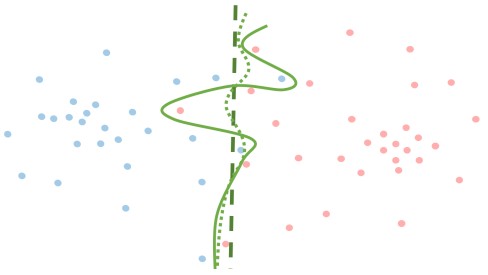

Figure 4: Motivation. The decision boundary between two classes (solid line) deviates from the optimal (dashed line). It tends to be strongly influenced by a few points, which can be reduced using NNs (dotted line).

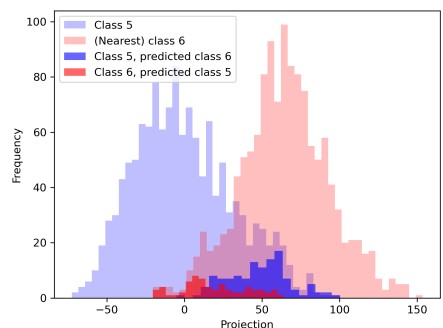

Figure 5: Distribution of projections (see Figure 2) for a ResNet-34 trained on Imagenet using the layer (outputs) prior to the last dense layer.

## 3 EVALUATION

Our evaluation focuses on image classification using various classifiers and datasets. We do the following:

- Assessing basic assumptions on distribution of layer activations (Section 3.2)
- Analyzing parameter sensitivity, e.g., impact of number of NNs and their weight on classifier accuracy (Section 3.3)
- Robustness to adversarial samples (Section 3.5) and label noise (Section 3.4)
- Performance of Algorithm LaSeNN for pre-trained classifiers (Section 3.6)

### 3.1 DATASETS, NETWORKS AND SETUP

Datasets used are CIFAR-10/100 Krizhevsky & Hinton (2009) (scaled to 32x32) and ImageNet Deng et al. (2009). As networks we used VGG Simonyan & Zisserman (2014), Resnet He et al. (2016), MobileNetv3 Howard et al. (2019) and ConvNextLiu et al. (2022) networks. We used pre-trained networks from the pytorch's torchvision library v0.15 based on ImageNetDeng et al. (2009) 'IMA-GENET1KV1' and trained multiple models on CIFAR-10/100 on our own. Training was standard, i.e., stochastic gradient descent with momentum 0.9, batchsize 128, weight decay of 0.0005, no data augmentation and 80 epochs (starting from learning rate 0.11 and decaying it twice by 0.1). We trained five networks for each configuration, i.e. hyperparameter setting and report the mean and standard deviation of metrics. We employ two common targeted adversarial attacks using the Pytorch advertorch library with default parameters and targets being set to "(index of ground truth class +1) modulo numberOfClasses". Specifically, we use the PGD attackKurakin et al. (2016) and a Basic Iterative Attack(BIA)Madry et al. (2018) which is an iterative extension of FGSMGoodfellow et al. (2014). If not stated differently, we use Algorithm 1 LaSeNN with the stated parameters in the algorithm. For VGG-13, we use as default, the third last convolutional layer for layer $i$, while for ResNet-10, we use the output of the second last 'BasicBlock'. Our own architecture variants as well as some additional code is part of the supplement.

### 3.2 DISTRIBUTION OF LAYER ACTIVATIONS

We aim to asssess our assumption that layer activations of most samples of one class are closer to each other than to those of other classes, or, put differently (activations) of class samples form dense clusters with dense centers that get increasingly sparse towards their boundary as illustrated through Figures 7 and 3, where we see a roughly Gaussian shape for the distribution of projections for each class. To verify this assumption, we compute for each point $X$ of the test set, the nearest neighbors

Table 2: Results for accuracy gains and adversarial Attacks on ImageNet

| Net | $corr(P, avgL2)$ | $samePred$ | $avgL2_{corr}$ | $avgL2_{wrong}$ | $avgL2_{change}$ |
|-----|------------------|------------|----------------|-----------------|------------------|
| ResNet-34 | -0.35 | 0.989 | 3.985 | 4.099 | 4.152 |
| CoNexT-Tiny | -0.49 | 0.993 | 1.434 | 1.565 | 1.663 |
| MobileNetv3-Large | -0.29 | 0.988 | 3.588 | 3.614 | 3.668 |

Table 3: Results for similarity metrics

| Net | Data | Metric | Acc. LaSeNN | Acc. Original | $\Delta$ Acc |
|-----|------|--------|-------------|---------------|--------------|
| ResNet-10 | Cifar-10 | L2 | $0.855_{\pm 0.003}$ | $0.854_{\pm 0.002}$ | $0.001_{\pm 0.001}$ |
| | | Cosine | $0.854_{\pm 0.002}$ | $0.852_{\pm 0.002}$ | $0.002_{\pm 0.0}$ |
| VGG13 | Cifar-10 | L2 | $0.819_{\pm 0.002}$ | $0.816_{\pm 0.001}$ | $0.003_{\pm 0.001}$ |
| | | Cosine | $0.825_{\pm 0.001}$ | $0.816_{\pm 0.001}$ | $0.008_{\pm 0.001}$ |
| ResNet-10 | Cifar-100 | L2 | $0.579_{\pm 0.001}$ | $0.574_{\pm 0.002}$ | $0.004_{\pm 0.001}$ |
| | | Cosine | $0.585_{\pm 0.003}$ | $0.575_{\pm 0.002}$ | $0.01_{\pm 0.002}$ |
| VGG13 | Cifar-100 | L2 | $0.511_{\pm 0.003}$ | $0.505_{\pm 0.002}$ | $0.006_{\pm 0.0}$ |
| | | Cosine | $0.522_{\pm 0.003}$ | $0.505_{\pm 0.002}$ | $0.017_{\pm 0.001}$ |

$NN_k(X)$ (for $k = 3$) in the training dataset and compute (1) pureness $P$: number of samples within $NN_k(X)$ that are of the same class as $X$ and (2) the average L2-distance $avgL2$ of the NNs to $X$[1].

If our assumption is correct, we expect that density measured by average $L2$ distance ($avgL2$) and pureness $P$ are negatively correlated $corr(P, avgL2) < 0$, i.e., higher density is expected for points of the same class (high pureness) and lower density for points of distinct classes(low pureness). As shown in Table 2 the Pearson correlation yielded values between -0.29 to -0.49 for all pretrained networks with p-values $< 0.001$. We also expect that most predictions remain unaltered due to using Algorithm LaSeNN, which is confirmed in Table 2 showing that more than 99% of all samples yield the same class prediction ($samePred$) if we compare the predictions of LaSeNN and the native classifier. We also expect that the mean distance to neighbors is lower for correctly classified points (since they are in dense areas near a center of a class with identical samples) than for incorrectly classified samples (since they are in sparser areas with samples of different classes), i.e., $avgL2_{corr} < avgL2_{wrong}$, which is also confirmed (see Table 2). We also expect that changes of class predictions due to LaSeNN occur primarily in low density areas (e.g. for large mean distances), i.e., $avgL2_{change} > avgL2$, which is also confirmed.

### 3.3 PARAMETER SENSITIVITY

First, we assess two common similarity metrics for high dimensional vectors: the (negative) $L2$-norm $sim(X, X') = -||X - X'||_2^2$ and cosine similarity $sim(X, X') = \text{cosine}(X, X')$. Our evaluation shows that both lead to gains for using NN but cosine leads to larger gains. This is expected since the space is relatively sparse (the Cifar 10/100 datasets are small with just 50k samples and the number of dimensions is large (at least 512 dimensions). In sparse spaces measures like cosine that neglect magnitude and are only concerned with direction are more favorable. In turn, L2 is more adequate for dense spaces, i.e., large training datasets. (We used L2 for our benchmarks with Imagenet.)

The outcomes for different layers $i$ are shown in Figure 4. Using deeper layer tends to lead to better results. Given a network of sufficient capacity after training all training samples will be perfectly classified, meaning they will have close to zero loss. In turn, the output of the very last layer is similar for all samples of a class, e.g., samples cannot be well discriminated using the final layer output.

In Table 5 we see that using NNs in addition to the sample to classify leads to gains from 0.1% up to about 3%. Gains are largest if the weight $w_q$ of the sample to predict is about 75% and that of the neighbors jointly only 25% though there is no strong sensitivity of the weight $w_q$. Using only NNs

---

[1]Distance to the kNN has been employed for density-based clustering, e.g., Schneider & Vlachos (2017)

Table 4: Results for layer $i$ used for similarity computation

| Net | Data | Layer $i$ | Acc. LaSeNN | Acc. Original | $\Delta$ Acc |
|---|---|---|---|---|---|
| ResNet-10 | Cifar-10 | prior to dense | $0.853_{\pm0.002}$ | $0.852_{\pm0.002}$ | $0.0_{\pm0.0}$ |
| | | prior to 4x4 pool | $0.854_{\pm0.002}$ | $0.852_{\pm0.002}$ | $0.002_{\pm0.0}$ |
| | | prior to last block | $0.864_{\pm0.004}$ | $0.854_{\pm0.002}$ | $0.01_{\pm0.002}$ |
| VGG13 | Cifar-10 | prior to dense | $0.817_{\pm0.001}$ | $0.816_{\pm0.001}$ | $0.001_{\pm0.0}$ |
| | | 2nd last conv | $0.816_{\pm0.001}$ | $0.816_{\pm0.001}$ | $-0.001_{\pm0.001}$ |
| | | 4th last conv | $0.825_{\pm0.001}$ | $0.816_{\pm0.001}$ | $0.008_{\pm0.001}$ |
| | | 6th last conv | $0.824_{\pm0.001}$ | $0.816_{\pm0.001}$ | $0.008_{\pm0.002}$ |
| ResNet-10 | Cifar-100 | prior to dense | $0.581_{\pm0.001}$ | $0.574_{\pm0.002}$ | $0.007_{\pm0.002}$ |
| | | prior to 4x4 pool | $0.585_{\pm0.003}$ | $0.575_{\pm0.002}$ | $0.01_{\pm0.002}$ |
| | | prior to last block | $0.593_{\pm0.001}$ | $0.574_{\pm0.002}$ | $0.019_{\pm0.001}$ |
| VGG13 | Cifar-100 | prior to dense | $0.508_{\pm0.002}$ | $0.505_{\pm0.002}$ | $0.003_{\pm0.001}$ |
| | | 2nd last conv | $0.518_{\pm0.002}$ | $0.505_{\pm0.002}$ | $0.012_{\pm0.0}$ |
| | | 4th last conv | $0.522_{\pm0.003}$ | $0.505_{\pm0.002}$ | $0.017_{\pm0.001}$ |
| | | 6th last conv | $0.518_{\pm0.003}$ | $0.505_{\pm0.002}$ | $0.013_{\pm0.001}$ |

Table 5: Results for weight $w_q$

| Net | Data | $w_q$ | Acc. LaSeNN | Acc. Original | $\Delta$ Acc |
|---|---|---|---|---|---|
| ResNet-10 | Cifar-10 | 0 | $0.853_{\pm0.0}$ | $0.852_{\pm0.0}$ | $0.001_{\pm0.0}$ |
| | | 0.52 | $0.854_{\pm0.0}$ | $0.852_{\pm0.0}$ | $0.001_{\pm0.0}$ |
| | | 0.76 | $0.854_{\pm0.0}$ | $0.852_{\pm0.0}$ | $0.002_{\pm0.0}$ |
| | | 0.88 | $0.854_{\pm0.002}$ | $0.852_{\pm0.002}$ | $0.002_{\pm0.0}$ |
| | | 0.94 | $0.854_{\pm0.0}$ | $0.852_{\pm0.0}$ | $0.002_{\pm0.0}$ |
| | | 0.97 | $0.853_{\pm0.0}$ | $0.852_{\pm0.0}$ | $0.001_{\pm0.0}$ |
| VGG13 | Cifar-10 | 0 | $0.799_{\pm0.002}$ | $0.816_{\pm0.001}$ | $-0.017_{\pm0.004}$ |
| | | 0.52 | $0.825_{\pm0.001}$ | $0.816_{\pm0.001}$ | $0.009_{\pm0.002}$ |
| | | 0.76 | $0.831_{\pm0.0}$ | $0.816_{\pm0.001}$ | $0.015_{\pm0.001}$ |
| | | 0.88 | $0.825_{\pm0.001}$ | $0.816_{\pm0.001}$ | $0.008_{\pm0.001}$ |
| | | 0.94 | $0.821_{\pm0.001}$ | $0.816_{\pm0.001}$ | $0.004_{\pm0.001}$ |
| | | 0.97 | $0.819_{\pm0.001}$ | $0.816_{\pm0.001}$ | $0.002_{\pm0.001}$ |
| ResNet-10 | Cifar-100 | 0 | $0.586_{\pm0.0}$ | $0.577_{\pm0.0}$ | $0.01_{\pm0.0}$ |
| | | 0.52 | $0.586_{\pm0.0}$ | $0.577_{\pm0.0}$ | $0.009_{\pm0.0}$ |
| | | 0.76 | $0.587_{\pm0.0}$ | $0.577_{\pm0.0}$ | $0.01_{\pm0.0}$ |
| | | 0.88 | $0.585_{\pm0.003}$ | $0.575_{\pm0.002}$ | $0.01_{\pm0.002}$ |
| | | 0.94 | $0.585_{\pm0.0}$ | $0.577_{\pm0.0}$ | $0.008_{\pm0.0}$ |
| | | 0.97 | $0.58_{\pm0.0}$ | $0.577_{\pm0.0}$ | $0.004_{\pm0.0}$ |
| VGG13 | Cifar-100 | 0 | $0.448_{\pm0.003}$ | $0.505_{\pm0.002}$ | $-0.058_{\pm0.004}$ |
| | | 0.52 | $0.525_{\pm0.002}$ | $0.505_{\pm0.002}$ | $0.02_{\pm0.002}$ |
| | | 0.76 | $0.536_{\pm0.002}$ | $0.505_{\pm0.002}$ | $0.031_{\pm0.002}$ |
| | | 0.88 | $0.522_{\pm0.003}$ | $0.505_{\pm0.002}$ | $0.017_{\pm0.001}$ |
| | | 0.94 | $0.515_{\pm0.004}$ | $0.505_{\pm0.002}$ | $0.01_{\pm0.002}$ |
| | | 0.97 | $0.511_{\pm0.002}$ | $0.505_{\pm0.002}$ | $0.005_{\pm0.0}$ |

(instead of the sample to classify) can be worse (i.e. for VGG13), but it can also be beneficial (i.e. for ResNet-10). For VGG13 it is worse. We believe that this is due to the nature of the latent space.

Considering the number of neighbors $k$ (Table 6, we see that improvements are largest, if just a single nearest neigbhor is used. This is not surprising, since the space is sparse and, thus, the larger $k$ the more dissimilar the neighbors are and the less valuable they are for prediction, i.e., they are more likely of another class than the ground truth class.

## 3.4 NOISY LABELS

Table 7 shows that using nearest neighbors leads to larger gains with growing noise, i.e., if we permute an increasing fraction of labels are permuted in the training data and the classifier is trained on this noisy data. This suggests that in latent space (induced by a classifier layer) training samples

Table 6: Results for number of nearest neighbors $k$

| Net | Data | $k$ | Acc. LaSeNN | Acc. Original | $\Delta$ Acc |
|---|---|---|---|---|---|
| ResNet-10 | Cifar-10 | 8 | 0.856±0.003 | 0.854±0.002 | 0.002±0.001 |
| | | 4 | 0.855±0.003 | 0.854±0.002 | 0.001±0.001 |
| | | 3 | 0.854±0.002 | 0.852±0.002 | 0.002±0.0 |
| | | 2 | 0.856±0.002 | 0.854±0.002 | 0.002±0.0 |
| | | 1 | 0.856±0.002 | 0.854±0.002 | 0.001±0.0 |
| VGG13 | Cifar-10 | 8 | 0.824±0.001 | 0.816±0.001 | 0.007±0.001 |
| | | 4 | 0.824±0.001 | 0.816±0.001 | 0.008±0.001 |
| | | 3 | 0.825±0.001 | 0.816±0.001 | 0.008±0.001 |
| | | 2 | 0.825±0.002 | 0.816±0.001 | 0.009±0.0 |
| | | 1 | 0.826±0.001 | 0.816±0.001 | 0.01±0.001 |
| ResNet-10 | Cifar-100 | 8 | 0.582±0.002 | 0.574±0.002 | 0.008±0.0 |
| | | 4 | 0.584±0.002 | 0.574±0.002 | 0.01±0.0 |
| | | 3 | 0.585±0.003 | 0.575±0.002 | 0.01±0.002 |
| | | 2 | 0.587±0.001 | 0.574±0.002 | 0.012±0.001 |
| | | 1 | 0.587±0.003 | 0.574±0.002 | 0.012±0.001 |
| VGG13 | Cifar-100 | 8 | 0.519±0.003 | 0.505±0.002 | 0.014±0.001 |
| | | 4 | 0.522±0.002 | 0.505±0.002 | 0.017±0.001 |
| | | 3 | 0.522±0.003 | 0.505±0.002 | 0.017±0.001 |
| | | 2 | 0.524±0.002 | 0.505±0.002 | 0.018±0.0 |
| | | 1 | 0.527±0.001 | 0.505±0.002 | 0.022±0.001 |

Table 7: Results for noisy labels

| Net | Data | Fraction permuted labels | Acc. LaSeNN | Acc. Original | $\Delta$ Acc |
|---|---|---|---|---|---|
| ResNet-10 | Cifar-10 | 0.0 | 0.854±0.002 | 0.852±0.002 | 0.002±0.0 |
| | | 0.01 | 0.841±0.0 | 0.839±0.001 | 0.002±0.0 |
| | | 0.04 | 0.807±0.0 | 0.807±0.001 | 0.0±0.0 |
| | | 0.08 | 0.77±0.0 | 0.766±0.0 | 0.003±0.001 |
| | | 0.16 | 0.708±0.002 | 0.703±0.001 | 0.005±0.001 |
| | | 0.32 | 0.59±0.003 | 0.581±0.003 | 0.01±0.0 |
| VGG13 | Cifar-10 | 0.0 | 0.825±0.001 | 0.816±0.001 | 0.008±0.001 |
| | | 0.01 | 0.814±0.003 | 0.806±0.004 | 0.008±0.001 |
| | | 0.04 | 0.796±0.001 | 0.783±0.002 | 0.013±0.002 |
| | | 0.08 | 0.767±0.004 | 0.753±0.004 | 0.014±0.0 |
| | | 0.16 | 0.716±0.001 | 0.696±0.001 | 0.02±0.0 |
| | | 0.32 | 0.604±0.0 | 0.577±0.003 | 0.026±0.003 |
| ResNet-10 | Cifar-100 | 0.0 | 0.585±0.003 | 0.575±0.002 | 0.01±0.002 |
| | | 0.01 | 0.576±0.0 | 0.566±0.0 | 0.01±0.0 |
| | | 0.04 | 0.542±0.001 | 0.529±0.001 | 0.013±0.001 |
| | | 0.08 | 0.502±0.002 | 0.49±0.002 | 0.012±0.0 |
| | | 0.16 | 0.437±0.0 | 0.424±0.002 | 0.013±0.002 |
| | | 0.32 | 0.332±0.002 | 0.315±0.002 | 0.017±0.0 |
| VGG13 | Cifar-100 | 0.0 | 0.522±0.003 | 0.505±0.002 | 0.017±0.001 |
| | | 0.01 | 0.52±0.001 | 0.501±0.001 | 0.019±0.002 |
| | | 0.04 | 0.493±0.002 | 0.473±0.003 | 0.02±0.001 |
| | | 0.08 | 0.47±0.002 | 0.45±0.002 | 0.02±0.001 |
| | | 0.16 | 0.422±0.0 | 0.399±0.001 | 0.022±0.001 |
| | | 0.32 | 0.33±0.01 | 0.307±0.007 | 0.024±0.003 |

with permuted (incorrect) label are still placed near samples of the correct label since they share similarities (beyond the class label).

## 3.5 ROBUSTNESS TO ADVERSARIAL ATTACKS

In Table 8 we see that the difference between LaSeNN and the unmodified classifier is larger for both of the targeted adversarial attacks indicating that our approach increases robustness to adversarial attacks. We believe that this is due to the fact that the adversarial samples are closer to the decision boundary and, thus, are more likely changed, when combined with NN.

## 3.6 PRETRAINED NETWORKS

While we have shown accuracy gains and robustness to adversarial samples on our self-trained small networks, it is unclear to what extent they also exist on large scale networks that are also trained using heavy data augmentation. To this end, we evaluate our technique on multiple pre-trained networks available through Pytorch's torchvision using the layer $i$ prior to the last dense layer,

Table 8: Results for adversarial attacks

| Net | Data | Attack | Acc. LaSeNN | Acc. Original | $\Delta$ Acc |
|---|---|---|---|---|---|
| ResNet-10 | Cifar-10 | None | 0.854±0.002 | 0.852±0.002 | 0.002±0.0 |
| | | BIA | 0.093±0.01 | 0.09±0.009 | 0.003±0.001 |
| | | PGD | 0.118±0.012 | 0.116±0.012 | 0.003±0.001 |
| VGG13 | Cifar-10 | None | 0.825±0.001 | 0.816±0.001 | 0.008±0.001 |
| | | BIA | 0.235±0.01 | 0.202±0.006 | 0.033±0.006 |
| | | PGD | 0.261±0.011 | 0.237±0.004 | 0.024±0.008 |
| ResNet-10 | Cifar-100 | None | 0.585±0.003 | 0.575±0.002 | 0.01±0.002 |
| | | BIA | 0.045±0.003 | 0.038±0.003 | 0.007±0.001 |
| | | PGD | 0.056±0.003 | 0.048±0.003 | 0.008±0.0 |
| VGG13 | Cifar-100 | None | 0.522±0.003 | 0.505±0.002 | 0.017±0.001 |
| | | BIA | 0.132±0.001 | 0.091±0.002 | 0.04±0.002 |
| | | PGD | 0.15±0.003 | 0.113±0.004 | 0.037±0.005 |

Table 9: Results for accuracy gains and adversarial attacks on pretrained networks on ImageNet

| Net | Attack | Acc. LaSeNN | Acc. Original | $\Delta$ Acc |
|---|---|---|---|---|
| ResNet34 | None | 0.7337 | 0.73316 | 0.00053 |
| | PGD | 0.01189 | 0.00991 | 0.00198 |
| | BIA | 0.01414 | 0.01191 | 0.00222 |
| ConvNext-Tiny | None | 0.82218 | 0.82128 | 0.00090 |
| | PGD | 0.01175 | 0.01035 | 0.00140 |
| | BIA | 0.01388 | 0.01252 | 0.00136 |
| MobileNetV3-Large | None | 0.74154 | 0.74056 | 0.00097 |
| | PGD | 0.00614 | 0.00506 | 0.00108 |
| | BIA | 0.00847 | 0.00707 | 0.00140 |

$w_q = 0.94$, and cosine similarity $sim(X, X') = \text{cosine}(X, X')$. We also employ augmentation for our nearest neighbor query, i.e., we compute the NNs for sample $X_q$ and for the horizontally flipped version $X'_q$ of sample $X_q$, but no other techniques. We take the union of the NNs (i.e. the original one and the flipped ones) and take those that are closest. In Table 9 we observe minor gains for all networks. This is somewhat surprising given that all these models are trained based on extensive data augmentation (e.g., random rotation, color jittering, random cropping and resizing, horizontal flipping), while our approach queries only leverage horizontal flipping. Aligned with our self-trained smaller networks we find that there is an increased robustness to adversarial attacks.

## 4 RELATED WORK

**Decision boundary**: Studying the decision boundary of neural networks dates back multiple decades Lee & Landgrebe (1997); Bishop (2006). Nowadays, studying the decision boundary is often motivated due to adversarial samples, which show that minor changes to a sample can result in crossing the decision boundary, e.g., deep learning networks are non-robust. Commonly, decision boundaries are also examined using measures and tools found in the context of adversarial examples, e.g., Ortiz-Jimenez et al. (2020); Karimi et al. (2019); Szegedy et al. (2014). Szegedy et al. (2014) discusses adversarial examples in deep learning, illustrating the sensitivity of decision boundaries in neural networks to slight input perturbations. Karimi et al. (2019) generates samples near the decision boundary based on techniques from adversarial samples and in a subsequent step they analyze the generated instances. Nguyen et al. (2015) presents the existence of "fooling" images—unrecognizable inputs that deep neural networks classify with high confidence, highlighting peculiarities in deep learning decision boundaries. Nguyen et al.'s findings shed light on the unusual and unexpected shapes that decision boundaries in deep networks can take. We approach decision boundaries more from the perspective that learnt representations are fixed and the task is to identify an optimal boundary separating samples. Ortiz-Jimenez et al. (2020) leverages tools from adversarial robustness to associate dataset features to the distance of samples to the decision boundary. In turn, they tweak the position of the training samples and measure the resulting changes on the boundaries. They show that deep learning networks exhibit a high invariance to non-discriminative features, and that the decision boundary of a neural networks only exist "as long as the classifier is trained with some features that hold them together"Ortiz-Jimenez et al. (2020). There are a num-

ber of theoretical and empirical findings on decision boundaries for neural networks not relying on ideas from adversarial samples. Fawzi et al. (2017) investigates topology of classification regions created by deep networks, as well as their associated decision boundary. The paper claims based on empirical evidence that regions (containing samples of a class) are connected and flat. Li et al. (2018) claims that the decision boundary of the last layer equals that of a hard SVM. Lei et al. (2022) measures the variability of the decision boundary. They show that the more variable the boundary, the less the networks generalizes. Recently, Mouton et al. (2023) predicts generalization performance based on input margins. That is, they use the variability computed based on PCA to assess generalization performance. Nar et al. (2018) argues that cross-entropy loss leads to poor margins, since samples can be very close to decision boundary. Support vector machines lead to better margins. In fact, years earlier this has been claimed empirically, i.e., Tang (2013) showed that using a margin-based loss instead of a cross-entropy loss can lead to improvements. Yang et al. (2020) states that thick decision boundaries lead to increased robustness. In the paper they propose training techniques to achieve this, but these techniques lead to significantly worse performance on the clean test sets and only improve on adversarial and out-of-distribution samples.

**Noisy Labels**: The impact of noise on decision boundaries cannot be understated. Noise in the training data can potentially lead to overfitting, manifesting as erratic decision boundaries Zhang et al. (2021). Large neural networks can "memorize" arbitrary noisy training dataZhang et al. (2021). However, noisy labels degenerates performance and research has investigated special techniques to deal with label noise. For example, Wu et al. (2020) constructs a topological filter to remove noisy samples. However, their approach falls short, when data is non-noisy and it is only shown to yield benefits if a large fraction of labels is noisy. Oyen et al. (2022) showed that label noise depends directly on feature space, i.e.,"when the noise distribution targets decision boundaries, classification robustness can drop off even at a small scale of noise."

**kNN**: Early works (prior to deep learning) Zhang et al. (2006) trained a SVM on NNs of a query sample. Theoretical works, e.g.,Cover (1968), studied also properties of neural networks. However, few theoretical and practical results are known relating deep learning and kNNs. Zhuang et al. (2020) designed a network for training a neural enforcing that a sample and its kNNs all belong to the same class based on a triplet loss. In contrast, we do not constrain training in any way, but rather compute NNs as they emerge by computing them based on the similarity of some layer activation of a trained classifier. Furthermore, our objective is to improve classifiers rather than primarily enforcing that decisions are based (solely) on kNNs.

**Memory and attention**: Our work also relates to works on including external memory in deep learning Graves et al. (2016) and to a lesser extent also attention, allowing to focus on specific (input) samplesVaswani et al. (2017) and Bahdanau et al. (2014). Our approach can be said to use training data as a read-only external memory in a static manner, in contrast to differentiable neural computersGraves et al. (2016) that allow read and write to memory and learn access. Attention allows to attend to (already processed) inputs within an input sequence. Our approach attends to specific training data used already for network training.

**Explainability**: Explaining using training data is common, e.g., influence functions Koh & Liang (2017) allow to explain the impact of training data on decisions. Naively, the influence of a training sample is computed by removing the training sample from the training data retraining the classifier on the reduced data, before assessing how the prediction for a specific sample changes. Our approach does not yield the influence but rather states that the output of a sample is determined by the NNs (at least with fraction determined by $w_q$). But it is then up to a human to compare the NNs and potentially assess concepts that are shared among them or to identify shared concepts using additional concept-based explainability methods, e.g. Schneider & Vlachos (2022).

## 5 CONCLUSIONS

While many approaches exist that target isolated problems such as interpretability, robustness against label noise or adversarial robustness, or better performance in general, we achieve with some extra computation"a little bit of most things" using a conceptual simple approach that also highlights that "overfitting" is a concern for deep learning on large datasets.

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

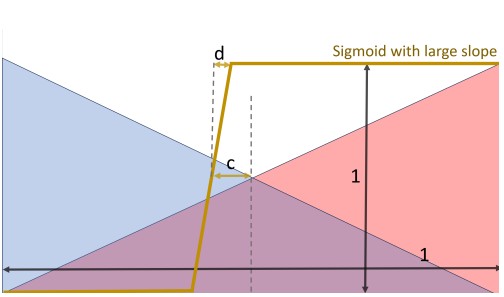

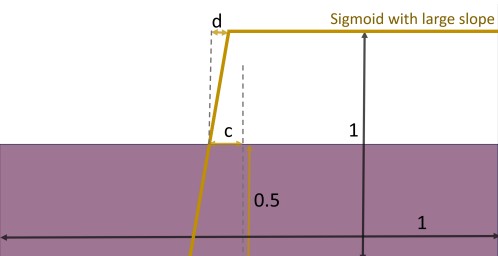

Figure 7: Uniform class probabilities highlighting the case, when class probabilities become more and more similar in a region, i.e., differentiating between classes becomes difficult in the region between two class centers. Note that further to the left or right (of the shown region in the figure) there might be cluster centers with only points of one or the other class. However, for a sigmoid activtion they have limited impact if their distance to th linear regime is far.

Figure 6: Unbalanced class probabilities as typical in real data, e.g. shown in Figures 7 and 3

## A    APPENDIX

## B    CONCEPTUAL UNDERSTANDING USING SIMPLIFIED MODEL

Let us get a better conceptual understanding of the benefits of using a simple classifier, i.e. a logistic regressor and NNs based on discussing two extreme class distributions of one dimensional data. The distributions are illustrated in Figures 6 and 7. That is, if we sample a point (for the training or test data) it is chosen according to these distributions, i.e., the class of a point $c(X)$ can be red (r) or blue (b). A point $X$ at position $c$ is blue with probability $p(c(X) = b|X = c) = 1 - c$ in Figure 6 (skewed triangular distribution) and $p(c(X) = b|X = c) = 0.5$ in Figure 7 (uniform distribution). The probability $p(X = c)$ to have a point $X$ at position $c \in [0, 1]$ is given by $p(X = c) = c$, i.e., if we add up the blue and red class distributions we obtain the uniform distribution for both Figures 6 and 7.[2].

A classifier $C$ outputs $p(c(X) = b)$. We consider three classifiers. A classifier $C_S$ using an simplification of a logistic regression, which has a small linear regime of width $2d$. A classifier $C_{NN}$ that is based on one NN. Classifier $C_{La}$ that outputs $C_{La}(X) = w_q C_S(X) + (1 - w_q) C_{kNN}(X)$ with $w_q > 0.5$, i.e., the sample $X$ to predict is most important as also assumed in Algorithm 1 and confirmed empirically.

The classifier $C_{NN}(X)$ simply returns $p(c(NN(X)) = b)$ according to distributions in Figures 6 and 7, i.e., the likelihood that the prediction of the nearest neighbor is blue or red. That is, we assume that we can more accurately estimate a point that is very close to the training data sample $NN(X)$ but there is still uncertainty, i.e., we do not predict the class with probability 1. This captures the idea that we do not use the label of the training data, but rather use the classifier output to make the prediction for the NN (as in Algorithm 1). But the classifier output is more accurate for the specific sample than possibly for a sample $X$ to infer.

The logistic classifier $C_S$ has a small linear region of width $2d$. That is $C_S(X) = p(c(X) = b) = 0$ if $X < c - d$, 1 if $X > c + d$ and $0.5(1 + (X - c)/d)$ otherwise as shown in Figures 6 and 7.

Let's consider the value of $c$ obtained when fitting $n$ data points. Say the training dataset is balanced (same number of samples for either class) and $c$ is chosen optimally so that there a minimal number of errors. The probability that a blue point $X$ is on one the side where red is more likely, i.e. at distance $0.5 + l$ is $p(X = 0.5 + l|c(X) = b) = 0.5 - l$. (Analogously for red) Thus, for some distance $l$, for $n$ points we expect $n \cdot (0.5 - l)$ to be blue and $n \cdot (0.5 + l)$ to be red. Due to the law

---

[2]This is done on purpose to keep the analysis straight forward and intuitive.

of large numbers the larger $n$ the less variation the mean exhibits and thus the less likely it becomes that there are more blue than red points at distance $l$ from the center $0.5$, implying that $c$ is expected to move closer to $0$ as $n$ grows. Say we obtained some classifier $C_s$ with some $c \neq 0$ for some training data of $n$ samples.

Let us consider $C_{La}$ and a query sample $X_q$. Outside the linear regime for $C_S$, i.e. $X_q \notin [c-d, c+d]$ the NN has no impact since the probability of blue is either 1 or 0 and the weight $w_q > 0.5$. That, is the predicted class is the same for $C_{La}$ as for $C_S$. If $X_q$ is chosen randomly from the distribution this happens with probability $1 - 2d$. Let's assume $X_q$ is close to the boundary, i.e., more specifically to simplify calculations directly at $1-c$ so that the classifier $C_S$ is undecided on whether to choose red or blue. Say there are $k$ red points to the left of $0.5-c$ and $n/2-k$ blue points with $k \leq n/2$. If points were distributed uniformly at random the chance to have the nearest neighbor to the left is larger $0.5$, since the density of points is larger to the left. However, to get a bound assume that we the $NN(x_q)$ is randomly chosen according to the distribution in Figures 6 and 7. Consider the distribution in Figure 6. Let us assume there is a nearest neighbor within distance $a$ so that $c-a > 0$ and $c+a < 1$.[3] Consider the probability that the $NN(X_q)$ is blue $p(NN(X) = b | NN(X_q) = x) = 1 - x$

$$p(NN(X_q) = b | NN(X_q) \in [c-a, c+a]) =$$

$$\frac{p(NN(X_q) = b | NN(X_q) \in [c-a, c+a])}{p(NN(X_q) = b | NN(X_q) \in [c-a, c+a]) + p(NN(X_q) = r | NN(X_q) \in [c-a, c+a]}$$

We have that
$$p(NN(X_q) = b | NN(X_q) \in [c-a, c+a]) =$$

$$\int_{x=0.5-c-d}^{0.5-c+d} 1 - x = 2(c+0.5)d$$

Similarly for $p(NN(X_q) = r | NN(X_q) \in [c-a, c+a]) = (1 - 2c)$. Plugging in yields

$$p(NN(X_q) = b | NN(X_q) \in [c-a, c+a]) =$$

$$\frac{2(c+0.5)d}{(1-2c)d + 2(c+0.5)d} = \frac{2c+1}{2} = 0.5 + c$$

Thus, we see that if the classifier $C_S$ is likely to incorrectly predict a sample as red due to the variation in training data leading to $c > 0$, the NN classifier $C_{NN}$ is likely to compensate, since the $NN(X_q)$ is more likely blue than red. Consider the distribution in Figure 7. Since samples of both classes are uniformly distributed, at any position $x \in [0,1]$ a point $X = x$ is as likely from the blue and red class. Thus using NN has no benefits. This illustrates that $NN$ are primarily helpful in areas, where the class densities are not well-balanced.

---

[3] $a$ depends on $n$. We have that the probability to have at least one point in $[c-a, c+a]$ being $1 - p(nopoint) = 1 - (1 - 2a)^n$.

