# OpenReview forum: "Improving classifier decision boundaries using nearest neighbors"
_ICLR.cc/2024/Conference — Submitted to ICLR 2024_

### Official Review · Reviewer_H15T · 2023-10-20

**Soundness:** 2 fair
**Presentation:** 2 fair
**Contribution:** 2 fair
**Rating:** 3
**Confidence:** 4

**Summary:**

This paper proposes a modified KNN method for evaluating the correctness of output of image classification models. Paper performs some experiments on CIFAR-10 and CIFAR-100 datasets, and concludes that the proposed method can provide “a little bit of most things” in terms of adversarial detection, accuracy, robustness to label noise, and interpretability.

**Strengths:**

Paper has a broad and practical view.

It proposes a relatively inexpensive algorithm with clear goals.

Overall, I find the approach interesting, and the method has the potential to become a useful contribution.

**Weaknesses:**

The main weaknesses, in my view, are the weak set of experiments and the weak literature review. There is also some arbitrariness in the method regarding the choice of layer and other parameters in the algorithm.

-------

The models used in the experiments have surprisingly weak accuracies. Paper spends considerable space describing the models, the dataset, and the specific details of how they have been trained. However, the reported testing accuracies are quite low. Paper can download standard models pretrained on any of the common datasets with 2 or 3 lines of code. The accuracy of such pretrained models, for example, the ones available on the pytorch library, or Hugging Face, are considerably higher than the ones this paper has trained. For example, the accuracy of models such as ResNet on the CIFAR-10 dataset are above 95%. The accuracy of ViT model on CIFAR-10 is around 98%. However, the accuracy of models used in this paper are about 85%. Am I reading the numbers correctly? I see the accuracy of ResNet on CIFAR10 is about 85% and the accuracy of ResNet on CIFAR-100 is 58%.

Because of this, the results do not provide much insight in my opinion. If authors were proposing some new training method that was compromising the testing accuracy for some other purpose such as adversarial robustness, that might have been understandable. But, the paper emphasizes that the proposed method does not need to make any modifications to the models, and it can be applied to any model. So, I don’t understand why experiments are performed on models with such low accuracies instead of pretrained models that are freely available.

The other shortcoming, in my view, is that the majority of results are on the CIFAR datasets. A full set of experiments on Imagenet would be more insightful.

--------

Choosing the layer seemed a bit arbitrary to me. I think it was not specified how one should choose the layer and how much the performance of the method would change if other layers are chosen. Authors probably have experimented with many layers and they are reporting the layer that was most effective? It would be useful if authors specify the variations of their results among the layers of the networks. It would also be useful to know how exactly authors have chosen those layers. Is that based on looking at the performance of the method on the testing set?

Another alternative is to just pick a specific layer and not change it among the experiments.

-------

It seems hard to understand Figures 6 and 7 as they don’t have axis labels, and they do not seem to be explained clearly enough. Also, Figures 6 and 7 seem to be going over the page limit.

----------

Literature review appears weak to me. There is no mention of literature on misclassification detection methods which directly relates to the paper’s claim that it can detect and reduce the mistakes of the models. Not only that literature needs to be mentioned, the results of this paper needs to be compared to those methods as well. One paper, as an example, from the misclassification detection literature:

-- Zhu, F., Cheng, Z., Zhang, X.Y. and Liu, C.L., 2023. OpenMix: Exploring Outlier Samples for Misclassification Detection. In Proceedings of the IEEE/CVF Conference on Computer Vision and Pattern Recognition (pp. 12074-12083).

Another paper that reviews that literature:

-- Jaeger, P.F., Lüth, C.T., Klein, L. and Bungert, T.J., 2022, September. A Call to Reflect on Evaluation Practices for Failure Detection in Image Classification. In The Eleventh International Conference on Learning Representations.

------

The references used in the paper for comparison are not recent and the paper misses the most recent methods published in the last few years. For example, in table 1, most of the references are from 2019 and 2020. These references may need to be updated.

Moreover, in table 1, there are no references assigned to adversarial robustness and accuracy. Perhaps this is a typo?

On the topic of adversarial detection, the paper does not use the most recent methods of detection. For example, the paper below uses a self-supervised embedding space and it reports that the detection of adversarial examples are significantly better in that space compared to the last hidden layer of trained image classification models. Authors would need to compare their method to the following method and possibly other competing methods.

-- Moayeri, M. and Feizi, S., 2021. Sample efficient detection and classification of adversarial attacks via self-supervised embeddings. In Proceedings of the IEEE/CVF international conference on computer vision (pp. 7677-7686).

The accuracy gains reported in Table 9 are negligible compared to the accuracy gains reported in the above paper.

If the author's method does not outperform some of the existing methods in the literature, that is okay in my view, because the proposed method is performing several tasks, and on the balance, it might be useful even if it does not outperform all the other methods. However, for each of the claims in the paper,  the paper needs to make comparisons with the most recent methods in the literature and demonstrate its capability, even if it is not as good.

-------

On the location of decision boundaries. The discussion around Figure 4 might be insightful, if authors go beyond a hypothetical decision boundary. My understanding from the paper is that Figure 4 is a hypothetical demonstration and not based on an actual model, is that right?

------------

What is demonstrated in figures 3 and 5 do not seem to imply any capability for detecting the mistakes. The mistakes are relatively close to the intersection of classes 5 and 6, however, a rather large portion of correct classifications are also at that same intersection shadowing all the mistakes of the model. When there is such a degree of mixture between correct and incorrect classifications, and the correct classifications are overwhelmingly more in numbers compared to the misclassifications, how can the method be helpful in improving the accuracy?

-------

Paper has missed the recent training methods that directly affect the decision boundaries. In particular, Mixup directly defines the location of decision boundaries in between the training samples. This has shown to have several advantages in adversarial robustness, accuracy, etc.

--Zhang, H., Cisse, M., Dauphin, Y.N. and Lopez-Paz, D., 2018, February. mixup: Beyond Empirical Risk Minimization. In International Conference on Learning Representations.

--Battleday RM, Peterson JC, Griffiths TL. Capturing human categorization of natural images by combining deep networks and cognitive models. Nature Communications. 2020.

Some of the discussions about the complexity of decision boundaries would not make much sense if the training method is mixup.

----------

I agree with the argument that using KNN provides some degree of interpretability. However, comparing testing samples to training samples has been a common approach. I think paper needs to go further than this and demonstrate, clearly and via examples, how that interpretability can be useful.

-------


Writing of the paper is weak, and at times, very ambiguous. For example, at the beginning of section 2, the paper states: “Second, the prediction is a combination of the network output of the sample to predict itself and its NNs, while for classical kNNs only the NNs are used to make a prediction”. I read this sentence a few times, and I don’t really get what it says. It is not clear to me what it means for “the sample to predict itself.”

Moreover, there are typos and grammatical errors.

Figure arrangements need improvement. Figure 4 comes two pages after it is discussed in the text. Figure 4 is discussed before figure 2. The arrangements seem premature and arbitrary.

**Questions:**

Please see questions and comments under weaknesses.

---

> ### Author Response · Authors · 2023-11-21
>
> As for reviewer N6Bu, this is really helpful feedback. While we naturally hoped for a better verdict, the reviewer provides solid arguments.

---

### Official Review · Reviewer_N6Bu · 2023-10-20

**Soundness:** 2 fair
**Presentation:** 2 fair
**Contribution:** 2 fair
**Rating:** 3
**Confidence:** 4

**Summary:**

The paper investigates how neural networks can be improved by integrating a k-nearest neighbors strategy in the class label computation. For this, given a data sample, the k-neighbors are determined among the training data samples. Thereby, the distance computations are performed in a latent space using a measure based on the Euclidean distance or the cosine similarity. Then, additionally to the given input, the class label for the determined k-neighbors is computed. Finally, the overall prediction is computed as a weighted average between the class label prediction for the original input sample and the predictions for the k-neighbors. With this approach, the not optimal decision boundaries of neural networks are improved leading to minor improvements in resistance to label noise, against adversarial attacks, classification accuracy, and to so+me degree interpretability. The proposed method is evaluated empirically.

**Strengths:**

1. The proposed method is simple and can be incorporated in any layered neural network used for classification. (originality)
2. The idea to consider the k-nearest neighbors to stabilize the prediction of a neural network is nice. (originality)
3. The measuring of the distances in the latent space where samples of one class should already be sufficiently close due to the feature engineering is good. (originality)
4. The evaluation of the method with respect to 4 different criteria is really good. (quality + originality)
5. Overall, I think that such a concept is interesting for the community. (significance)
6. The paper is well-written and has a good structure. (clarity)

**Weaknesses:**

Please see for further information Questions where I will explain these points in more detail.

1. some claims/statements in the paper are not sufficiently backed up with results or references. (quality)
2. the proposed method should be better discussed with respect to state of the art. (quality + originality)
3. The improvements are minor (as honestly stated by the authors). In general, IMHO, this is not a problem but the authors could have presented the results in a better way. For instance, they could have performed significance test to check if the differences are statistically significant. (quality)
4. Some design choices in the evaluation pipeline are nor properly motivated or justified.

**Questions:**

I recommend that the authors read these questions carefully to improve the quality of the submission. If these concerns/questions can be clarified, I will reassess my scoring of the paper.

1. The first sentence in the paper ends with "optimal decision boundaries." Throughout the paper this statement is repeated. However, it is not convincingly explained or references what an optimal decision boundary is. So my question is what is an optimal decision boundary? I noted that the authors state "optimal decision boundary <==> optimal classifier." But in this case what is an optimal classifier? Right now, this term of optimal decision boundary is the major motivation for the approach. So if this term could be clearly defined maybe the authors could show mathematically why the proposed method works (beyond the simple example presented in Appendix B).
2. Could the authors elaborate on the following, IMHO, similar work?
   - "Deep k-Nearest Neighbors: Towards Confident, Interpretable and Robust Deep Learning" by Nicholas Papernot and Patrick McDaniel;
   - The concept of randomized smoothing is similar because it averages the prediction too (even if the neighbors are sampled from a distribution). For example, see "Certified Adversarial Robustness via Randomized Smoothing" by Cohen, Jeremy and Rosenfeld, Elan and Kolter, Zico.
3. "In our work, we show that classifiers do generally not learn optimal decision boundaries (also) due to the fact that these boundaries lie in areas of few training samples and thus naturally suffer from over-fitting." The claim that they suffer from over-fitting is not sufficiently backed up by results. So I recommend that the authors either present convincing results for this claim or reference to appropriate work. Could the authors comment on that?
4. I wonder if the measure illustrated in Figure 2 is a good measure. The problem is that the measure will become incorrect if one class encapsulates the other (like everything is class A but there is a circle somewhere which is class B). Could this be a problem for the results derived?
5. Section 3.1: What is the motivation for computing an adversarial attack for "ground truth class + 1? Common is that the closest adversarial sample is approximated (no matter of the class label). How could this design choice affect the results?
6. Table 2: The caption says "Results for ... adversarial attacks." Which column corresponds to the results of the adversarial attacks?
7. I recommend that the authors discuss the significance of the results in terms of a significance test. Right now, the improvements are really minor and it is difficult to assess if these changes are significant. Maybe the authors could use the Brunner-Munzel test. Could the authors report such results?
8. Table 5: How were the steps for the hyperparameter tuning selected? It seems quite counter intuitive.
9. "We believe that this is due to the nature of the latent space." Could the authors present evidence/results for this guessing? Maybe a t-SNE plot, UMAP could help here.
10. Table 6: Best results for k=1. This is really surprising for me because I would expect that the power of the method lies in the averaging over several k. However, with k=1 showing the best results, I wouldn't consider this as a proper averaging. The statement that this is due to a sparse space is not convincing. How can the authors conclude that the space is sparse?

***

Some notes/comments to the authors, which I have not considered for the scoring:

1. Please have your paper proofread by an English native speaker
2. There are several typographical errors in the paper. For instance, at several places there are missing blanks: "samplesGoodfellow et al. (2014)", "nearest neighbors(NNs)". The first Equation in Section 2 has no punctuation.
3. Section 3.3: "shown in Figure 4" should be Table 4.
4. I wouldn't call a dataset with 50k samples small.
5. I find the abbreviation NN rather confusing as this is the common abbreviation for neural network.
6. Is the learning rate stated (0.11) really correct? From my experience, this sounds pretty high.

**Details Of Ethics Concerns:**

None.

---

> ### Author Response · Authors · 2023-11-21
>
> Thank you for your constructive feedback. While we naturally disagree with the overall valuation, the feedback is really helpful and also adequate.

---

### Official Review · Reviewer_RTTt · 2023-11-02

**Soundness:** 2 fair
**Presentation:** 1 poor
**Contribution:** 1 poor
**Rating:** 1
**Confidence:** 4

**Summary:**

The paper proposes to use nearest neighbors in the latent space of neural networks to improve their accuracy, improve their robustness to adversarial samples and provide some form of interpretability of their decisions.

**Strengths:**

- The proposal of analyzing the latent space through the lens of nearest neighbors makes sense and can provide some insight on model behavior.
- Results reported show small, although noticeable performance improvement over the baseline models.

**Weaknesses:**

- The overall writing and presentation is poor. In particular, it is difficult to make sense of many figures (e.g., Figs. 2, 3 and 5). Algorithm 1 is also quite messy, while not carrying much information.
- The claims are broad and not well-supported (see Table 1). For instance, the interpretability claim is not accurate nor properly illustrated with experiments. Being able to identify the nearest neighbors and use them to interpret classification of the current sample is a local and quite weak way to interpret the decision, we don’t really know how the decision is made for the current sample. This is far from being explainable AI. Likewise, the robustness claims are not well supported, with no comparison with state-of-the-art methods made for improving robustness (e.g. adversarial training).
- All results are reported on rather old and small backbones, that is ResNet-10 and VGG13, on CIFAR-10 and CIFAR-100. Baseline results are low, that is 85.4%/81.6% on CIFAR-10 with ResNet-10/VGG13, and 57.4%/50.5% on CIFAR-100 with the same backbones. Results with common backbones with are rather at 95.5% for CIFAR-10/ResNet-18 (https://paperswithcode.com/sota/image-classification-on-cifar-10) and 81.1% for CIFAR-100/Wide ResNet (https://paperswithcode.com/sota/image-classification-on-cifar-100). It is easy to show gains when starting with poor baselines, and the improvements are rather thin – less than 1% on CIFAR-10 and around 1-2% for CIFAR-100 (ref Table 6).
- The state-of-the-art is not very well developed, rather throwing ideas and discussing of too many aspects. The idea of using instance-based classification in feature space is not new and has been explored before. As relevant work, I point to Wang and Sabuncu 2022 (https://openreview.net/forum?id=iEq6lhG4O3) in TMLR, which present a much more convincing work around similar ideas (minus the adversarial aspects, which are not well supported in the current paper).

**Questions:**

I don’t have specific questions as I think the paper presents outstanding issues that need to be solved regarding the paper presentation, motivation, and experiments. See section weaknesses for the details.

**Details Of Ethics Concerns:**

No ethics concerns with this paper.

---

> ### Author Response · Authors · 2023-11-21
>
> Thank you for the feedback.
> > All results are reported on rather old and small backbones
>
> No, you missed Table 9. 3  -> We provide networks that were not trained by us on Imagenet and can be considered state-of-the-art (they are from Pytorch)
>
> > Being able to identify the nearest neighbors and use them to interpret classification of the current sample is a local and quite weak way to interpret the decision, we don’t really know how the decision is made for the current sample.
>
> This is exactly what also the paper does the reviewer mentioned. Also the concept of influence functions which explains decision based on individual training samples is common. Sample-based explanations might not be a final solution to XAI, but  it is well-established.
>
> > the interpretability claim is not accurate nor properly illustrated with experiments
>
> We agree that its presentation should be improved and experiments are helpful, but the idea here is similar to the original kNN idea (also present in the cited paper by the reviewer). We give the sample to predict a weight of 0.88, the rest comes from NN. Thus a fraction of 0.12 is well-explainable (given the output of the (last) linear layer) is used, i.e., one can look at the sample and its class (sample-based / influence based explanation). The work cited by the reviewer only uses NN to classify, which is equivalent to classical kNN. It makes their work more explainable than ours, but their work falls short in other ways (performance, complexity of implementation...)
>
> > Likewise, the robustness claims are not well supported, with no comparison with state-of-the-art methods made for improving robustness (e.g. adversarial training).
>
> We only claim that we can improve robustness at no major disadvanges (aside from computation). We don't claim that our approach improves other approaches that only focus on robustness (possibly trading other aspects in favor of it)

---

### Meta-Review · Area_Chair_MnBt · 2023-12-10

**Metareview:**

The paper proposes to apply nearest neighbors in the latent space of pretrained neural networks to improve their accuracy, interpretability, robustness to adversarial samples and label noise, and to produce clear decision boundaries. The idea is simple and elegant, and empirically produces improvement in some benchmarks.

However, this feels like a paper from circa 2010. It misses lots of literature that study clustering in latent space. The methods it compares to as benchmarks are also quite outdated. All reviewers voted unequivocally reject. We encourage the authors to read up on the literature and try to improve the method. We look forward to see future submissions with improved ideas and results.

**Justification For Why Not Higher Score:**

The idea is reasonable, however the experiments failed to compare to more modern architectures. The overall writing and presentation can also be improved.

**Justification For Why Not Lower Score:**

N/A

---

### Decision · Program_Chairs · 2024-01-16

Reject